# Meibomian Gland Dysfunction: What Have Animal Models Taught Us?

**DOI:** 10.3390/ijms21228822

**Published:** 2020-11-21

**Authors:** Mingxia Sun, Isabel Y. Moreno, Michelle Dang, Vivien J. Coulson-Thomas

**Affiliations:** College of Optometry, University of Houston, Houston, TX 77204, USA; msun9@central.uh.edu (M.S.); iymoreno@cougarnet.uh.edu (I.Y.M.); mmdang2@cougarnet.uh.edu (M.D.)

**Keywords:** Meibomian gland, animal models, dry eye disease

## Abstract

Studies have estimated that currently 344 million people worldwide and 16.4 million adults in the US have some form of dry eye disease (DED). It is believed that approximately 70% of DED cases are due to some form of evaporative dry eye, for which Meibomian gland dysfunction (MGD) is the major cause. Unfortunately, currently there is no effective treatment for MGD, and solely palliative care is available. Given the importance of MGD in DED, there has been a growing interest in studying Meibomian gland development, homeostasis and pathology, and, also, in developing therapies for treating and/or preventing MGD. For such, animal models have shown to be a vital tool. Much of what is known today about the Meibomian gland and MGD was learnt from these important animal models. In particular, canine and rabbit models have been essential for studying the physiopathology and progression of DED, and the mouse model, which includes different knockout strains, has enabled the identification of specific pathways potentially involved in MGD. Herein, we provide a bibliographic review on the various animal models that have been used to study Meibomian gland development, Meibomian gland homeostasis and MGD, primarily focusing on publications between 2000 and 2020.

## 1. Introduction

In epidemiological studies performed globally, the prevalence of dry eye disease (DED) is estimated to range from 5% to 50%. DED affects 6.8% of the United States adult population (approximately 16.4 million people) and is the most commonly reported condition in eye care clinics [1]. There are primarily two types of DED that are categorized by their etiology: aqueous-deficient DED and evaporative DED (Figure 1). Studies have shown that up to 87% of dry eye patients have evaporative DED, for which the major cause is Meibomian gland dysfunction (MGD). The Meibomian gland is a specialized sebaceous gland, which is embedded in the superior and inferior tarsal plates of the eyelids and produces a lipid secretion called meibum. The action of blinking causes Meibomian glands to secrete meibum, and, thereafter, evenly spreads this meibum across the ocular surface. This thin layer of meibum lies over the aqueous layer, protecting it from evaporation and thereby stabilizing the tear film. The tear film is vital in maintaining corneal homeostasis, nourishing and protecting corneal and conjunctival epithelial cells. Any functional changes to the Meibomian gland that alter its meibum production affect tear film stability, and, ultimately, disturb corneal homeostasis, leading to epithelial damage. Consequently, patients with MGD will report symptoms of eye irritation, such as ocular pain and discomfort, and will also show signs of inflammation [2,3].

The term MGD encompasses various functional abnormalities of the Meibomian gland. There are three types of MGD divided into two categories: low delivery and high delivery. Low delivery is associated with hyposecretory or obstructive MGD, while high delivery pertains to hypersecretory MGD. Hyposecretory MGD involves decreased meibum secretion resulting from abnormal Meibomian gland function without orifice blockage, while obstructive MGD, the most common type of MGD, results from ductal obstruction. High delivery MGD is characterized by hypersecretory Meibomian glands, where a large volume of lipids is excreted onto the surface of the lid margin when pressure is applied to the tarsal plate. Common treatments for MGD include maintaining eyelid hygiene, warm compresses, and anti-inflammatory therapies [4]. Studies have suggested that MGD starts with the occlusion of terminal ducts, which leads to a cascade of changes starting with the accumulation of meibum within collecting ducts, causing a build-up of pressure and ultimately leading to the atrophy and loss of Meibomian glands [5]. Hyperkeratinization has been suggested to be a major cause of obstructive MGD. To that effect, hyperkeratinization of the ductal epithelium can lead to ductal occlusion, which in turn leads to cystic dilation of the duct and, ultimately, results in gland drop out [6]. Recently, some speculation has been raised that hyperkeratinization is a major contributor of MGD and research is ongoing to substantiate this [6,7,8,9,10]. Recent studies have hypothesized that aging is a potential cause for MGD, often referred to as age-related MGD (ARMGD). Research has suggested that ARMGD is caused primarily by a decline in the number of differentiating meibocytes, which ultimately leads to Meibomian gland dropout and abnormal lipid excretion [6].

Age and gender are considered to be the greatest risk factors for DED. Progressive Meibomian gland loss (drop-out) occurs with age, which is accompanied by reduced quality and quantity of meibum [11,12,13,14,15]. A cross-sectional study involving 177 subjects aged 21–93 years found that eyelid margin and Meibomian gland abnormalities are significantly associated with age, while no significant association was found between Meibomian gland-related changes and gender [11]. However, in a cross-sectional study conducted in New Zealand, females were found to be significantly more likely to be affected by DED, MGD and asymptomatic ocular surface disease [16]. A separate study reported increased changes in Meibomian gland morphology in elderly men when compared to women in an Asian population [17]. In 2017, The Tear Film and Ocular Surface Society reported that there was a lack of specific data on the impact of gender and age on MGD. Thus, currently, further studies are required to establish a possible link between gender and MGD [1].

Currently, the pathogenesis of MGD and ARMGD remains largely unknown and is a developing field of research. Importantly, understanding the pathogenesis of MGD and ARMGD is necessary for developing treatment strategies. Given the recent increased recognition of the importance of MGD, there has been a growing trend towards studying Meibomian gland development, homeostasis and pathology, and developing therapies for treating the different forms of MGD. Animal models have played an integral role in our understanding of the underlying mechanisms that cause this complicated disease. Importantly, many animals, such as dogs, also suffer from debilitating DED that can often lead to loss of vision and require veterinary interventions. The primary shortcoming with current animal models for fully understanding the scope of the disease is that they do not replicate the entire pathophysiology of human MGD. Consequently, it is also important to understand the limitations of each model. In this review, we will summarize the major findings regarding Meibomian gland and MGD research and highlight relevant anatomical and pathological features of MGD in different animal models and how they relate to humans. We will also describe the various animal models that have been used to study MGD and ARMGD, including their benefits and limitations. We will also outline the advances that have been made in the field based on findings from studies using different transgenic mouse strains.

## 2. Meibomian Gland Structure and Function

Meibomian glands are modified holocrine sebaceous glands embedded within the upper and lower eyelids [6]. The ductal system is lined with a basal layer of proliferating stratified squamous epithelial cells and attached to the ducts are acini, bundled structures containing secretory cells which are connected to the central duct through smaller ductules. The secretory cells, named meibocytes, are located towards the center of the acinus and secrete meibum into the ductal system. The acini are lined with proliferating basal cells, and, as these cells divide, they move in towards the center of the gland, differentiate into meibocytes and begin to accumulate lipids. When the hyper mature meibocytes reach the center of the acini, the nucleus becomes pyknotic and they undergo apoptosis, releasing their contents, the meibum, into the ductules. Thus, basal cells in the periphery of the acini continuously proliferate to produce meibocytes that move centripetally towards the center of the gland as they mature. When meibocytes reach the center of the gland they disintegrate, releasing their contents into the collecting ductules. From the ductules, the meibum moves towards the central duct and through the orifice onto the ocular surface. Studies have shown that new meibocytes have an average cell cycle of 4.1 days and take approximately 9 days from when they first move into the gland from the basal layer to when they complete maturation and are shed into the collecting duct [5].

As part of the holocrine secretory process, there is a continuous loss of meibocytes paired with their constant regeneration. The basal cells of the Meibomian gland continuously proliferate throughout life to supply new cells. Multiple groups have speculated over the years that, as in most tissues, Meibomian glands must contain a pool of quiescent progenitor cells continuously supplying new basal cells over time. Previous studies by Parfitt et al. [8] show that, in mice, quiescent progenitor cells exist at the terminal ends of the ductal epithelium within Meibomian glands, and it is further speculated that these “stem cell-like” cells are necessary for long-term maintenance of viable basal cells. The authors suggest that the loss of these progenitor cells in the aging population could play a role in the pathogenesis of MGD [8]. Olami et al. [18] also investigated the location of potential stem cells within the Meibomian gland and found stem cell-like populations residing around the circumference of acini, with transient amplifying cells existing in the basal layer of acini.

## 3. Meibomian Gland Morphogenesis

As previously stated, Meibomian glands are modified holocrine sebaceous glands and, therefore, have similar development to sebaceous glands [6]. They grow during the third to seventh month of gestation from the mesoderm [5]. After lens placode invagination from the lens vesicle, the mesodermal loose connective tissue in the lid folds, differentiating into the tarsal plate, and the mesenchymal tissue morphs into the eye folds, creating the palpebral aperture [19]. The epithelial cord of the Meibomian anlage forms the ductules and acini, and it has been observed that lipid production by the Meibomian anlage leads to differentiation of the upper and lower lids during the seventh month of gestation [5]. Mouse Meibomian gland development is very similar to that of humans and starts at embryonic day (E) 18.5 with the formation of the epithelial placode at the fused eyelid margin epithelium [20,21]. Epithelial invagination into the eyelid mesenchyme happens between post-natal day (P)1 and P3, during which the epithelial cord forms and lengthens. At P8, the ductal and acinar structures begin to differentiate and, at P15, normal Meibomian gland morphology can be seen [20].

## 4. Meibomian Gland Dysfunction and Pathology

Studies have suggested that MGD starts with the occlusion of terminal ducts, which leads to a cascade of changes starting with the accumulation of meibum within collecting ducts, causing a build-up of pressure and ultimately leading to the atrophy and loss of Meibomian glands [5]. Hyperkeratinization has been suggested as one of the causes of obstructive MGD. This is due to the fact that hyperkeratinization of the ductal epithelium can lead to ductal occlusion, which in turn leads to cystic dilation of the duct and, ultimately, results in gland drop out [6]. Currently, some forms of palliative care are designed to remove and prevent orifice blockage, and include warm compresses, maintaining lid hygiene, lubricants and antibiotic ointments or topical steroids for severe cases [5]. However, unfortunately there is no established standard care for MGD, and treatments therefore vary depending on practitioners [4]. There is a general consensus that the accumulation of meibum within Meibomian glands and changes in the composition of meibum play an important role in MGD. The accumulation of meibum within Meibomian glands can lead to changes in its composition. Elegant studies have demonstrated that the lipid composition of meibum directly affects its physical properties [22,23,24,25,26,27]. Additionally, an increase in protein accumulation has been shown to increase the viscosity of meibum during MGD [5,28]. An increase in the viscosity of meibum decreases its flow through the ductules and collecting duct, which, in turn, increases pressure within the gland. A study conducted by Jester and colleagues [6] examined the composition of meibum through stimulated Raman scattering (SRS) microscopy. They found that acinar regions of the Meibomian gland have a lower protein concentration than ductal regions, suggesting that meibum undergoes maturation during the secretory process, with proteins detaching from the lipids. Desiccating stress has been shown to increase the acinar turnover rate, which results in proteins not being detached from the lipids during the secretory process, and this leads to an increase in viscosity and, consequently, an increase in pressure within the gland, which ultimately causes clogging of the duct [6,28]. The microbiota of the ocular surface, and potentially even within the Meibomian gland, could also play a role in MGD. Bacteria at the lid margin have been shown to secrete enzymes that can alter the composition of meibum, and these changes can increase the viscosity of meibum, irritate the epithelia and induce keratinization [5,29]. As mentioned above, the loss of progenitor cells has also been proposed as a cause for ARMGD. Oxidative stress is known to cause DED, whereby reactive oxygen species (ROS) cause tissue damage in Meibomian glands and lacrimal glands, inducing inflammation and decreasing lipid secretion leading to MGD [30]. Corneal neuropathies can also cause dry eye. After laser-assisted in situ keratomileusis (LASIK), about 20–40% of patients develop chronic DED 6 months after surgery, and Denoyer et al. [31] describe how LASIK-induced dry eye occurs mainly due to the breakdown of subbasal nerves. As more research on MGD is conducted with special emphasis on why it occurs, there has been a shift in focus away from hyperkeratinization and ductal occlusion to, instead, focus mostly on age-related changes to the Meibomian gland and ARMGD. In ARMGD, it is believed there is primarily a decline in meibocyte differentiation and lipid synthesis resulting in gland dropout and abnormal lipid secretion.

Although a lot is known about MGD, currently the exact pathogenesis of MGD remains elusive. There are likely to be multiple causes of MGD (many of which remain unknown), and these could potentially vary from person to person, and thus, more importantly, would require different forms of treatment. It is therefore vital to (1) understand the exact pathogenesis of MGD, (2) categorize different forms of MGD based on their pathogenesis, (3) be able to easily diagnose the different forms of MGD based on their pathogenesis, and, finally, (4) design appropriate treatment options for each form of MGD.

## 5. Pathogenesis of MGD in Humans and Other Mammalian Species

Animal models can provide critical information to better understand human diseases and especially contribute to the advance in therapeutics. When studying MGD, an ideal animal model must include some, ideally most, of the following pathological features of the human disease, (1) obstruction of the glands, (2) keratinization of the ducts, (3) dilation of the ductules, (4) atrophy of the glands, (5) eventual gland drop-out, and (6) hyperemia or fibrosis of the eyelid margin. As in human MGD, the entire lids should be diffusely affected in a chronic and progressive manner. Depending on the purpose of the study, animal models could be used to study spontaneous MGD, similar to what is observed in humans, or experimental MGD animal models could be designed to induce this condition.

Canines have a longer lifespan when compared to traditional lab animals, for example rabbits, mice and rats, and, interestingly, many canine breeds are predisposed to DEDs. In fact, Marta Viñas et al. [32] reported that MGD is a common condition in dogs. To date, studies have been unable to demonstrate an association between MGD and skull conformation, even though dry eye is a common condition in brachycephalic breeds [32]. In a recent case report, Yasunari Kitamura et al. [33] performed a histopathologic analysis of Meibomian gland dropout in a 14-year-old Cairn terrier and the findings were quite similar to human MGD dropout. General ophthalmologic examination revealed abnormal ocular surface characterized with chronic inflammation signs including eyelid swelling, mild hyperemia of conjunctiva, pigment glaucoma and dense vascularization with corneal ulcers [33]. Hematoxylin–eosin (HE) staining of resected eyelid tissue revealed the absence of normal gland lobular and ductal structures at the dropout site, and noncontact infrared meibography (NIM) revealed Meibomian gland atrophy [33]. Another case report demonstrated that a two-year-old neutered female mongrel dog presented sebaceous adenitis and concurrent MGD [34]. This condition is also common in human patients. Thus, dogs are an excellent model to study MGD and test different potential therapies. The use of dogs in experimental research is highly controversial, requires specialized facilities, and can be very expensive; however, clinical research in collaboration with veterinary clinics studying the progress of DED in pets can be very valuable. Importantly, progress in this field would also benefit dogs suffering from DED and MGD, which often results in loss of vision, and, normally, pet owners are open to experimental therapies that could potentially improve their pet’s condition.

Other species have also been explored for studying MGD and age-related dry eye diseases, such as the rabbit and nonhuman primates. Various animal models of MGD have demonstrated abnormal keratinization of the Meibomian gland, including the rabbit epinephrine-induced MGD, the primate polychlorinated biphenyl-induced MGD, and the rhino mouse genetic MGD models. When studying signs of experimental induced DED, the rabbit model seems to be a more suitable animal model when compared to rodents (rats and mice), primarily because rabbit eyes offer better accessibility for ocular surface evaluations. Surgical removal of the lacrimal gland in rabbits has been shown to induce aqueous deficiency DED [35]. Moreover, orchiectomy and ovariectomy in rabbits have been used to study the influence of hormones on the structure and function of the lacrimal and Meibomian glands [36,37]. Interestingly, the Butovich group found that androgen stimulates Meibomian gland genes associated with lipid metabolic pathways, and also found that androgens prevent keratinization [37]. However, in our literature searches we were unable to find reports on ARMGD in rabbits.

The mouse model is the most widely used animal model in research, and mouse studies have contributed immensely to the understanding of MGD and DED. Currently, the mouse model is the most widely used model to study the mechanism of MGD and DED. The mouse model also offers a diversity of knockout and transgenic strains that enable the investigation of the contribution of particular pathways or cell types to MGD and DED. Moreover, as outlined below, lipidomic and genomic studies in mice and humans have revealed that these two species are very close in terms of biochemistry and physiology of their meibum, further supporting the mouse model as a reliable and convenient animal model to study human MGD [22,25]. Despite the vast number of mouse studies that have been dedicated towards studying the Meibomian gland, MGD and DED, there have been few dedicated towards studying ARMGD [38,39]. Importantly, recent studies have shown that C57BL/6 and BALB/c mice naturally present ARMGD, with evident Meibomian gland dropout from 1 year of age, suggesting that the mouse model is also a valuable tool for studying ARMGD [39,40].

## 6. Interspecies Variations in Meibum Composition

Meibum is the lipid-rich secretion that is made by the Meibomian gland and deposited onto the ocular surface and has been identified as very diverse and unique in nature. Its composition differs from other lipid pools throughout the human body and presents interspecies variability [35]. The precise composition of the lipids within the meibum is vital for maintaining tear film stability and even slight changes may lead to clinical manifestations of DED [41].

Over the years, the meibum composition of many different animal species (such as the steer, gerbil, rat, rabbit, dog, cat, cow, hamster, and mouse) has been analyzed using different approaches. These studies showed that the lipid composition of meibum throughout the different animal species does not replicate that of human meibum. Harvey and colleagues [25] used GC/MS to characterize the lipid composition of mouse meibum. This technique hydrolyzes and then methylates and/or trimethylsilylates the meibum prior to analysis, thus, only structures resulting from free sterols, fatty alcohols, and fatty acids are evaluated, with a limited capability to analyze underivatized wax esters. This study identified cholesterol as a major constituent of mouse meibum [25]. Intact wax esters were shown to be composed of branched-chain alcohols and both branched-chain and unsaturated acids. Butovich and colleagues [22] compared Meibomian lipidomes of three commonly used animal models, specifically, dogs, mice, and rabbits, to that of humans, and found that mice followed by dogs were the closest species to humans in terms of meibum lipid composition. The rabbit meibum lipidome, on the other hand, was very different to the lipidomes of all other species tested [22].

The composition of human meibum has been thoroughly characterized in recent years, with particular focus on lipids. Human meibum is composed of a complex mix of lipids, including “nonpolar” lipids (65–70%), wax esters (WE—~41%), cholesteryl esters (CE—~31%), free cholesterol (Chl—<0.5%), triacylglycerols (TAG—~1%), and smaller amounts of more polar compounds (~30%), such as free fatty acids (FFA—~0.1%), phospholipids (PL—~0.05%), sphingomyelins, ceramides (Cer), and other lipids [23,26,41]. Nicolaides and colleagues [27] found that fatty acids and fatty alcohols found in complex lipids of meibum have very long chains and extremely long chains, and are often branched. Recently, over one hundred major individual lipid species belonging to more than a dozen lipid classes have been identified in the meibum of humans and animal models [41]. The defining feature of all these major classes of complex Meibomian gland lipids is the chain length (ranging from C32 to C36), such as alcohol chains in wax esters, fatty acids in cholesteryl esters, ω-hydroxy fatty acids in (O)-acylated ω-hydroxy fatty acids, and α,ω-diols in diacylated α,ω-diols. Free fatty acids with very long chains and extremely long chains have also been reported in meibum [42,43].

Amphiphilic lipids (often called “polar lipids”, ~4.8%), including (O)-acylated ω-hydroxy fatty acids, free cholesterol, free fatty acids, phospholipids and ceramides, are known to be present in healthy human meibum, but their function remains to be established. Recently, extremely long-chain (O)-acylated ω-hydroxy fatty acids were identified in meibum (at approximately 3% (w/w) of healthy human meibum), and this novel amphiphilic lipid was suggested to play an important role in stabilizing the tear film by forming an interphase between the lipid and aqueous sublayers. Although (O)-acylated ω-hydroxy fatty acids have also been identified in canine, rabbit and mouse meibum, there are structural variations. Specifically, the most abundant (O)-acylated ω-hydroxy fatty acid homolog in human and canine meibum has a ω-hydroxy-C32:1-FA moiety, while in rabbits it has a ω-hydroxy-C32:0-FA moiety, and in mice it has a ω-hydroxy-C34:2-FA moiety [22,26,44,45]. Diacylated α,ω-diol lipids have also been reported to be a significant component of meibum, making up approximately 3.6% of total lipids [27]. In fact, diacylated α,ω-diols dominate the mass spectrum of rabbit meibum and are also prominent components of canine meibum, but are less preponderant in human and mouse meibum [45]. A variation in the degree of diacylated α,ω-diol saturation has also been noted among different species, changing from predominantly fully saturated lipids, i.e., without any unsaturated fatty acids or fatty alcohol chains in rabbits, to predominantly mono- and di-unsaturated fatty acids and fatty alcohol chains in humans and mice. The resemblance between the lipid composition of human and mouse meibum supports the use of mice for MGD and ARMGD studies.

## 7. Mouse Models Used to Study DED and ARMGD

Meibomian gland function is essential for maintaining ocular surface health and homeostasis. Any functional changes to Meibomian glands can lead to abnormal meibum secretion and/or altered lipid composition. MGD has been recognized as a leading cause of DED, but the pathophysiology of MGD and ARMGD remain largely elusive. Over the years, various animal models have been designed that are able to recapitulate some of the pathophysiological alterations seen in human MGD. Herein, we will outline the advances made in the fields of DED and MGD based on findings obtained using mainly experimental mouse models. In particular, we will discuss the different models that have been established to study different aspects of MGD and DED, including the advantages and disadvantages of each model. We will also discuss the different transgenic mouse models that have been used to unveil the contribution of different molecular pathways and cell types to Meibomian gland development, Meibomian gland homeostasis and MGD. Recently, many transgenic and genetic mutant mouse models have been generated that develop anatomical features similar to MGD and ARMGD, providing valuable insights into molecular pathways and cell types involved in Meibomian gland morphogenesis and homeostasis, and potential causes of MGD. We also summarize the different genetic mouse models (Table 1) that have been used to study different aspects of the Meibomian gland published from 2000 to 2020.

### 7.1. Aging Mice as a Model for ARMGD

For many years, the aging process has been associated with MGD in humans and other mammalian species [46,47,48]. Interestingly, a body of evidence exists suggesting that mice also develop ARMGD with a similar pathogenesis to the human disease, with Meibomian gland atrophy already evident at 1 year of age [40]. Jester and colleagues showed a 70% decrease in the average cross-sectional gland area in aging mice (12 months) [39] compared to young mice (2 months) using Metamorph [39,49]. This same group further demonstrated that aged mice (2 years old) show significantly fewer acini when compared to younger mice (2 months old), specifically a 27.1% decrease in total gland volume associated with a 54.4% decrease in lipid volume. For this study, a nonlinear optical (NLO) array tomography was used to reconstruct the mouse Meibomian gland using digital image processing software [49,50,51]. Recently, a novel mouse transillumination meibography device was used to assess the Meibomian glands of young (3 months old or less) and aged (over 15 months old) BALB/c mice [52]. This study found there was a significant decrease in Meibomian gland density in aged mice when compared to young mice [52]. A recent report by Yoon and collaborators further confirmed aging C57BL/6 mice as a valuable tool for studying ARMGD [40]. This study analyzed aging and aged male mice (one and two years old, respectively). They found evidence of dry eye and related ocular surface damage at one year of age, although they noted that significant atrophy of Meibomian glands was only evident after two years of age [40]. They also observed that oxidative stress was significantly increased in both lacrimal glands and Meibomian glands after two years of age but did not find significant stem cell senescence in either gland [40]. Further studies have demonstrated that Meibomian glands in older mice undergo age-related atrophy associated with decreased acinar cell proliferation, differentiation and lipid synthesis [9]. These changes take place in concert with changes in PPARγ expression and localization, suggesting that PPARγ is a master regulator of meibocyte differentiation and function. Further on we discuss the role of PPARγ in ARMGD in more detail. In summary, simply aging mice can serve as a valuable animal model for studying ARMGD.

### 7.2. Correlation between Hormone Levels and MGD

Although studies indicate that there is a close association between gender and MGD, evidence is still lacking regarding the role of the endocrine system. Importantly, aging is associated with a loss of sex hormones in both men (andropause) and women (menopause). Many epidemiological studies have shown that MGD and DED often occur during menopause, placing the female gender as a risk factor for DED [1]. However, as mentioned above, the mechanism behind the sex-associated predisposition to DED remains unknown. Sullivan and colleagues [36,53,54] found that Meibomian glands of male and female rats, rabbits and humans are potential targets for androgen hormones, identifying that they express androgen receptor mRNA, androgen receptor protein within acinar epithelial cell nuclei and mRNA for 5α-reductases types 1 and 2. Additionally, androgen signaling has been shown to regulate the expression of numerous genes in rabbit and mouse Meibomian glands [36,55]. Moreover, androgen hormones have been shown to control the quality and quantity of lipids produced by the Meibomian gland, thereby, promoting the formation of a stable lipid layer in the tear film [53]. Androgen insufficiency occurs with aging and, thus, as mentioned above, is believed to be involved in ARMGD [12].

The growth hormone (GH) is a peptide hormone that stimulates growth, cell reproduction and cell regeneration in humans and other animals [56,57]. The GH/insulin-like growth factor 1 (IGF-1) axis, which mediates endocrine, paracrine and autocrine pathways, is a primary driving force of mammalian growth and plays an integral role in aging in multiple species [56,58]. GH levels also decline with age, and this decrease is believed to contribute towards ARMGD [57]. Sullivan and colleagues [57,59] found that IGF-1 activates the PI3K/AKT and forkhead box O1 signaling pathways, stimulating human Meibomian gland epithelial cell proliferation, increasing the expression of sterol regulatory element-binding protein, and promoting lipid accumulation. Based on these findings, this group further investigated the role of the GH/IGF-1 axis in Meibomian gland homeostasis, finding that GH-deficient mice present striking morphological alterations. Specifically, mice with decreased GH or complete loss of GH (knockout mice) present significantly smaller Meibomian glands with hyperkeratinized and thickened ducts, secretory acini located within duct walls, and poorly differentiated acini (atrophy). In contrast, mice that up-regulate GH present significantly larger Meibomian glands when compared to control mice [60]. Thus, these findings suggest that the GH/IGF-1 axis could be a valuable target for treating ARMGD.

### 7.3. Role of PPARγ in MGD and ARMGD

In addition to the effects of hormones, the altered expression of peroxisome proliferator-activated receptor gamma (PPARγ) has been identified as another contributing factor to aspects of ARMGD [9]. The PPARs are members of the nuclear receptor superfamily of ligand-inducible transcription factors. In mammals, there are three PPARs: PPARα (also called NR1C1), PPARβ/δ (also called NR1C2) and PPARγ (also called NR1C3) [61]. By binding to PPAR-responsive regulatory elements as obligate heterodimers with retinoid X receptor (RXR), the PPARs control the expression of gene networks involved in adipogenesis, lipid metabolism, inflammation, and maintenance of metabolic homeostasis [62,63]. Among the three PPAR families, PPARγ is most highly expressed in white adipose tissue (WAT) and brown adipose tissue (BAT), where it is a master regulator of adipogenesis, as well as a potent modulator of whole-body lipid metabolism and insulin sensitivity [62,64]. PPARγ signaling networks are extremely complex with tissue specific regulatory pathways [65]. Tissue-specific PPARγ knockout mice have been crucial in helping dissect the relative contributions of PPARγ to Meibomian gland morphogenesis and the pathogenesis of MGD [65]. In particular, Jester and colleagues [7,66] have made significant contributions to unveiling the role of PPARγ in Meibomian gland development, meibum secretion and MGD. There is a marked change from a cytoplasmic (2–6 months of age) to nuclear localization of PPARγ during aging (1–2 years of age), which is associated with a significant decrease in meibocyte cell cycling and Meibomian gland size in mice [7,51]. Moreover, cytoplasmic PPARγ localization in younger mice has been associated with multiple small lipid droplets throughout the cytoplasm, while in aged mice the altered PPARγ subcellular localization was correlated with a decrease in lipid producing acinar tissue [7]. This trans-location pattern was also found in a 5- and 7-month-old *ApoE^−/−^* MGD mouse model in concert with Meibomian gland dropout [39]. PPARγ expression is first detected in mouse eyelids at P3 and localized to the inner portion of the invaginating epithelial cord, prior to the formation of acinar structures [20]. Oil-red-O staining was used to detect lipids at this stage of Meibomian gland development, and lipids were identified within the inner cord co-localizing with cells expressing PPARγ [20]. As Meibomian gland morphogenesis progressed, PPARγ expression was lost in the ductal epithelium and became limited specifically to the acinar compartment, both in cytoplasmic vesicles and nuclei of acinar cells [20]. These findings indicate that PPARγ is expressed by lipid synthesizing cells (meibocytes) in mature Meibomian glands, but not in ductal epithelial cells [20,67]. Thus, PPARγ could be used as a potential marker for meibocyte differentiation [7,61,68]. As mentioned above, in humans, Meibomian gland dropout has been tightly correlated with patient age and has also been correlated with nuclear PPARγ staining [9]. Further studies using conditional inducible loss of PPARγ in the Meibomian gland could further shed light on its contributions towards Meibomian gland homeostasis and ARMGD.

### 7.4. Mouse Models Used to Study the Role of Oxidative Stress in MGD

Aging is associated with an accumulation of reactive oxygen species (ROS) that damage all components of cells, including proteins, lipids and DNA, leading to age-related chronic diseases [69]. The term “oxidative stress” refers to an imbalance between high levels of ROS and low cellular antioxidant defenses [70]. Superoxide dismutases (SODs), the first line of defense against oxygen free radicals in organisms, can limit the potential toxicity of these molecules and control broad aspects of cellular life. There are three classes of SODs: Cu/Zn SODs (SOD1), Mn SOD/Fe SODs (SOD2), and Ni SODs (SOD3). Among them, SOD1 is widely distributed in tissues and represents 90% of total SOD activity [71]. SOD1 is a secreted enzyme that catalyzes the conversion of superoxide radicals (O_2_^-^) to hydrogen peroxide, which can then be further reduced to water, and therefore, has an important role in the defense against oxidative stress in tissues [72]. SOD1-deficient mice (*Sod1^−/−^* mice) exhibit high levels of oxidative stress when compared to wild-type mice, and have been shown to display ocular defects, including age-related macular degeneration and lacrimal gland dysfunction, as observed in humans [73,74]. To further test if more tissues are impacted, Kojima and colleagues examined the morphological features and secretory functions of lacrimal glands in SOD1 knockout mice (*Sod1^−/−^* mice). *Sod1^−/−^* mice at 50 weeks of age present obvious lacrimal gland interlobular fibrosis (atrophy), which was associated with significantly increased infiltration of CD45+, CD4-, CD11b and Gr-1-positive inflammatory cells when compared to age-matched control mice. *Sod1^−/−^* mice present a significant decrease in tear quantity accompanied by increased tear instability and corneal epithelial damage at 50 weeks of age when compared to age-matched control mice. These mice also present a significant increase in IL-6 and TNF-α levels within the tear film as early as 10 weeks of age when compared to control mice. This same group also investigated whether *Sod1^−/−^* mice exhibit any Meibomian gland alterations [75]. Histological analysis of the Meibomian glands of *Sod1^−/−^* mice showed decreased acinar density with larger lipid droplets, an increase in CD45^+^ inflammatory cell infiltration and fibrosis in comparison to control mice. Moreover, 50-week-old *Sod1^−/−^* mice exhibit a significant increase in positive TUNEL staining within Meibomian glands when compared to control mice. In addition, extensive oxidative stress-related lipid (4-HNE staining, lipid peroxidation) and DNA damage (8-OHdG DNA oxidative stress marker) were observed in Meibomian gland epithelia, which increased with aging from 10 to 50 weeks of age in both *Sod1^−/−^* and control mice. Lipid and DNA damage appeared to be more extensive in 50-week-old *Sod1^−/−^* mice when compared to age-matched control mice. These results suggest that *Sod1^−/−^* mice suffer many of the biochemical and structural changes found in human MGD, suggesting a role for oxidative stress in this pathology. Thus, SOD1-knockout (*Sod1^−/−^* ) mice have been suggested as a useful animal model for studying various aspects of ARMGD.

### 7.5. Cautery-Induced Obstruction of Meibomian Gland Orifices and MGD

Obstructive MGD is a leading cause of DED among clinical patients, but the pathological mechanisms leading to obstruction remain unclear. Obstructed Meibomian glands are believed to be caused by hyperkeratinization of the excretory duct and/or increased viscosity of the secretion (meibum) with subsequent deficiency of the tear film lipid layer. Animal models have been developed in an attempt to experimentally induce MGD caused by obstructed glands. In 1989, Gilbard et al. reported that closing Meibomian gland orifices of rabbits by “light cauterization” increased tear film osmolarity and produced ocular surface damage, including decreased conjunctival goblet cells and corneal epithelial glycogen levels, similar to what is observed by cauterizing the lacrimal gland excretory duct (keratoconjunctivitis sicca [KCS]) [76]. Essentially, cauterization causes keratinization of the exterior portion of the Meibomian gland opening, which leads to blockage of the Meibomian gland orifice by scarring and also duct obstruction, meibum stasis and cystic dilation, which ultimately lead to MGD (acinar atrophy and gland dropout). An early study demonstrated that MGD is induced successfully in mice 12 weeks after cauterization due to the induction of fibrosis of the orifice, suggesting that glandular changes can occur secondary to obstruction [77]. Electrocauterization was also used to cause mechanical damage to Meibomian gland orifices in order to generate an MGD rat model [78]. For such, under a slit lamp microscope, a slim metal guidewire was inserted (0.25 mm in diameter) 1–2 mm into the upper and lower Meibomian gland orifices and the guidewire was fulgurated for 0.1 s to destroy part of the Meibomian gland orifice. A hoary secretion was observed in the Meibomian gland opening at four weeks post-surgery, and the rats gradually developed Meibomian gland atrophy and obvious signs of corneal surface damage, all in line with the clinical manifestations of MGD. All these techniques used damage/scarring to the orifice as a means to induce permanent loss of the Meibomian gland, similar to “gland dropout” caused by obstructive MGD observed in the clinic.

### 7.6. Desiccating Stress Induces Dry Eye-Like Symptoms and Associated Meibomian Gland Functional Changes

For many years the most common cause of MGD was believed to be epithelial hyperkeratinization of the collecting duct, which leads to gland duct obstruction, meibum stasis, cystic dilation, and eventually disuse acinar atrophy and gland dropout [5,48,79]. More recent studies have demonstrated that reduced basal cell proliferation contributes towards MGD [8,10,67]. The risk factors for MGD that affect meibocyte differentiation and renewal can be either intrinsic (e.g., aging) or extrinsic (e.g., environmental stress). Desiccating stress, a common denomination for harsh dry environmental conditions, can lead to tear film abnormalities, including tear hyperosmolarity, which leads to ocular surface damage [80]. So far, the mouse model for desiccating stress-induced inflammatory dry eye is the only available animal model to study the underlying mechanisms of environmental DED [81,82]. This model was used to study the effects of desiccating stress on Meibomian gland function [28]. Changes in meibum within the acinar/ductule regions were investigated using stimulated Raman scattering (SRS) microscopy to measure the ratio of protein and lipid [83]. The results show that low humidity stress causes a significant increase in basal acinar cell proliferation in untreated mice after 5- and 10-day exposure. Since the Meibomian gland is a holocrine gland, in which cells undergo differentiation and disintegration to release lipids, increased proliferation suggests an increase in meibocyte differentiation and, consequently, increased disintegration and release of lipids into the Meibomian gland duct. Dilation of the duct was also noted, which could be secondary to the increased release of meibum. Dilation of the duct could also be secondary to an increase in meibum viscosity caused by inadequate removal of proteins from lipids under desiccating stress conditions [84]. Based on these findings, the authors concluded that retention of proteins in meibum may lead to a more rigid lipid tear film layer that is subject to fracture and instability, potentially explaining some of the clinical parameters of MGD, including tear film instability.

### 7.7. Association between Dyslipidemia and MGD

In recent years, several clinical studies have demonstrated that moderate-to-severe MGD was associated with high total cholesterol and low-density lipoprotein blood levels [85,86]. Thereafter, dyslipidemic animal models were used to study the Meibomian gland and, more specifically, MGD [87,88,89]. To investigate whether systemic lipid disorders can be associated with DED and/or MGD, a study used three transgenic adult mice models, apolipoprotein-E knockout (APOE-KO), low-density lipoprotein receptor knockout (LDLR-KO) and a mouse model with overexpression of human apolipoprotein CIII (ApoCIIIKI), compared to age- and gender-matched C57BL/6 mice [87]. Surprisingly, these hyperlipidemic mice subjected to a high-fat diet did not present any signs of DED [87]. Thus, the authors concluded after this study that DED is not caused exclusively by dyslipidemia, and instead must be dependent on a combination of genetic and environmental conditions [87]. Apolipoprotein E (ApoE) is a component of plasma lipoproteins and serves as a ligand for cell-surface lipoprotein receptors such as low-density lipoprotein (LDL)-receptor (LDLR), LDLR related proteins, very low-density lipoprotein (VLDL), and high-density lipoprotein (HDL) [90]. Apolipoprotein E knockout (*ApoE^−/−^*) mice are characterized by a marked increase in total plasma cholesterol levels and develop microvasculature lesions; making *ApoE^−/−^* mice a possible model for studying the mechanisms of hyperlipidemia and atherosclerosis in MGD and DED [91]. Recently, Bu et al. [89] found that *ApoE^−/−^* mice present hypertrophic eyelids at 5 and 7 months of age, which was accompanied by Meibomian gland dropout and disordered acini and ducts in both upper and lower eyelids, when compared to age-matched control mice. These mice also present punctate corneal staining and signs of corneal damage, all clinical features of MGD [89]. Histological analysis of *ApoE^−/−^* mouse eyelids revealed plugging of the Meibomian gland orifice, duct dilation and heteromorphic acinar morphology. Taken together, this study of *ApoE^−/−^* mice demonstrated that obstructive MGD and hyperlipidemia are closely associated [89]. Furthermore, oral polyunsaturated omega-6 fatty acid supplements, specifically linoleic and gamma-linolenic acid, were shown to be beneficial in the treatment of MGD [92]. Thus, supplementing the diet with certain lipid-based supplements could be beneficial in the prevention and/or treatment of MGD. In contrast, a study using HR-1 hairless mice fed a limited lipid diet (HR-AD, a special diet with limited lipid content) developed atopic dermatitis-like symptoms characterized by severely dry skin and dermal inflammatory cell infiltration [93]. This mouse model was also used to study the eyelids and, more specifically, Meibomian glands. For such, HR-1 hairless mice were fed a HR-AD diet for up to 16 weeks. After 11 weeks, the mice presented marked posterior blepharitis around the eyelid margin, plugging orifices, and toothpaste-like meibum, which were not observed in HR-1 mice receiving normal diet [94]. Therefore, a HR-AD diet was enough to trigger an ocular phenotype that resembles human MGD and could potentially be used in future clinical studies. Most recently, Wu et al. used high-fat diet (60 kcal% fat, HFD) and successfully induced dry-eye-like ocular surface damages in a C57BL/6 mouse model, including decreased tear production, notable Oregon green dextran (OGD) staining, distinct conjunctival goblet cell loss and squamous metaplasia in the ocular surface [95]. Taken together, these studies suggest that the lipid composition in an individual’s daily diet may also contribute to the development of MGD.

### 7.8. Role of Ectodysplasin A (EDA) in the Meibomian Gland and “Tabby” Mice

The X-linked anhidrotic-hypohidrotic ectodermal dysplasia mouse (Tabby) is a naturally developed ectodysplasin A (EDA) mutant mouse strain [96]. These mice present abnormalities in skin appendages and teeth development, and lack Meibomian glands [97]. Therefore, this strain represents an extreme MGD condition in which no meibum is spread onto the ocular surface. The changes in ocular surface phenotype of Tabby mice include corneal epithelial defects, keratitis, corneal ulceration, neovascularization, keratinization, blepharitis, and conjunctivitis in the late stages from 12 to 36 weeks of age [97]. The results indicate that during the early interval from 6 to 8 weeks changes were already occurring similar to those seen in evaporative DED [97]. This group found that aqueous tear secretion remained relatively stable with only a transient increase at 10 weeks of age [98]. Lacrimal gland hematoxylin and eosin staining of 8-week-old mice showed no obvious structural difference between wild-type mice and Tabby mice. In addition, no obvious difference was found in goblet cell density between 4- and 8-week-old mice in these two groups [98]. Thus, Tabby mice are a useful model for studying contributions of the Meibomian gland to DED, and also for studying the effects of MGD on the ocular surface and lacrimal gland.

### 7.9. FGFR2 Signaling in Meibomian Gland Homeostasis and Potential Role in MGD

Fibroblast growth factor (FGF) signaling is involved in a variety of biological processes, including cell growth, migration, differentiation, survival, and apoptosis [99]. It is also essential for embryonic and neural development and adult tissue homoeostasis [100,101,102]. The environment into which FGF is secreted influences FGF function [102]. An in vivo mouse genetic study showed that conditional gene deletion of Fgfr2b, an FGF receptor 2 (FGR2) isoform, in epidermal cells causes striking abnormalities in hair and sebaceous glands without affecting animal survival [103]. Reneker and colleagues [104] used a triple transgenic mouse strain (Krt14-rtTA; tetO-Cre; *Fgfr2^flox/flox^*), referred to as *Fgfr2^CKO^* mice, to study the role of FGFR2 in Meibomian gland homeostasis. After a 2-week induction period, these mice developed severe Meibomian gland atrophy associated with reduced lipid (meibum) production (Oil-red-O staining), as well as evaporative DED symptoms related to MGD in human subjects [42,79]. These symptoms included ocular irritation, loss of corneal luster and macerated eyelids. This study suggests that FGFR2 signaling plays an essential role in maintaining acinar basal cell proliferation in Meibomian glands. Thus, FGFR2 activity could be required for the maintenance of Meibomian gland acinar progenitor/stem cell population, which is critical for Meibomian gland maintenance and homeostasis in adult mice [104].

### 7.10. Gene Disruption of Fatty Acid (FA) Synthesis and MGD

Meibum, the lipid layer of the tear film, is integral to maintaining a stable tear film. Qualitative and/or quantitative changes in the meibum lead to evaporative dry eye [5,105]. The majority of meibum lipids consists of nonpolar lipids, including cholesteryl esters (~65% in humans), wax esters (~25%) and triacylglycerols (~4%), minor polar lipids and others. Meibum cholesteryl esters and wax esters contain iso- or anteiso-branched fatty acids/alcohols, which are classified into long-chain (C11–C20) or very long-chain (>C20) fatty acids/alcohols. (O)-acylated ω-hydroxy fatty acids predominantly contain monounsaturated C30–C34 ω-hydroxy FAs and are uniquely present in meibum. It was identified that fatty acid elongases are responsible for the synthesis of very long-chain fatty acids. Seven elongases (elongation of very long-chain fatty acids (ELOVL)) [1,2,3,4,5,6,7] exist in mammals, and each exhibits characteristic substrate specificity towards acyl-CoAs [106]. So far, three ELOVL mutant mice have been generated—specifically, *Elovl*1, *Elovl3* and *Elovl4* knockout mice.

ELOVL1 is involved in the synthesis of very long-chain acyl moieties and acts on saturated C18:0- to C26:0-CoAs and monounsaturated C18:1- to C22:1-CoAs [106]. To overcome conventional *Elovl1^−/−^* neonatal lethality, Sassa and colleagues [107] generated *Elovl1^−/−^ Tg(IVL-Elovl1)* mice to study the role of Elovl1 dependent very long-chain meibum lipids in DED. They found that cholesteryl esters with saturated very long-chain fatty acids of C25:0–C27:0 were largely reduced (4–11% decrease compared to control mice) in *Elovl1^−/−^ Tg(IVL-Elovl1)* mice. Elovl1 deletion leads to the clear shortening of saturated fatty acid/alcohol moieties in two major meibum lipids: cholesteryl esters and wax esters. On the other hand, they found that monounsaturated fatty acid/alcohol moieties are reduced in cholesteryl esters, wax esters and (O)-acylated ω-hydroxy fatty acids in mutant mice. Evaporative dry eye signs were observed in these mice, including reluctant eye opening, frequent eye blinking and excessive tearing, thus confirming an important role for *Elovl1* in meibum production. With aging, ELOVL1 deficiency leads to corneal damage (41% in 5–11 months of age, and 88% in ≥12 months of age).

The fatty acid elongase ELOVL3 is involved in the synthesis of C20-C24 saturated and monounsaturated very long-chain fatty acids [108]. As seen with ELOVL1 deficient mice, Butovich and colleagues found that ELOVL3 protein ablation impacts mouse ocular surface physiological features and Meibomian glands [108]. ELOVL3-ablated (*E3hom*) mice show delayed eye opening, weeping eyes, crusty eyelids, eyelid edema, highly vascularized cornea and tarsal plates, slit eye and increased tearing, all resembling symptoms observed in human DED [108].

The enzyme ELOVL4 is required for the synthesis of extremely long-chain fatty acid residues (longer than C28), which are found in the nonpolar fraction of tear film lipids, such as cholesteryl esters of (O)-acylated ω-hydroxy fatty acids, wax esters and cholesteryl esters [109]. Mice carrying mutations in *Elovl4* (heterozygous Stgd3 mice) have increased eyelid blink rates, partially open eyes, protruding Meibomian gland (MG) orifices and swollen eyelids, with anatomical changes in the Meibomian gland [110]. These data indicate that ELOVL4 plays a role in lipid biosynthesis and is important for maintaining Meibomian gland and ocular surface homeostasis.

Meibum cholesteryl esters and wax esters contain iso- or anteiso-branched fatty acids/alcohols at unusually high levels. The conversion of fatty acids to fatty alcohols is required for fatty acyl-CoA reductases [111]. Recently, as part of the Jackson Laboratory’s Knockout Mouse Project (KOMP), mice lacking fatty acyl CoA reductase, named 2Far2tm2b (KOMP) Wtsi/2J (hereafter referred to as *Far2^−/−^*) mice, were used as an animal model for deciphering the complex pathogenesis of human primary cicatricial alopecia [112]. Interestingly, significant abnormalities were noted in both the sebaceous glands and Meibomian glands of *Far2^−/−^* mice [113]. Since the sebaceous gland is a holocrine gland, sebaceous gland cells (sebocytes) mature as they move into the center of the gland gradually accumulating uniformly sized vacuoles containing clear lipids, which are eventually released as sebocytes disintegrate into the duct [114]. In *Far2^−/−^* mice, sebocytes have very few clear cytoplasmic vacuoles, and those that are present vary in size but overall are much smaller than those in normal mice. Sebocytes in these mice do not show signs of maturation throughout sebaceous glands, and do not fully disintegrate as they enter the sebaceous gland duct. Mitochondria appear swollen and occasionally contain small, solitary, electron dense granules, which is not observed in control tissues. The Meibomian glands of *Far2^−/−^* mice show markedly dilated ducts leading to the surface of the mucocutaneous junction, which itself is markedly ectatic [113].

### 7.11. Role of NF-κB in Meibomian Gland Development

Hypohidrotic ectodermal dysplasia (HED) is a human syndrome defined by the maldevelopment of one or more ectodermal-derived tissues, including the epidermis and cutaneous appendices, teeth, and exocrine glands [115,116]. Kappa-enhancer in B cells (NF-κB) has been suggested as the key molecular mechanism that causes HED [117]. In 2003, Cascallana et al. [118] reported that ectoderm-targeted transgenic mice overexpressing glucocorticoid receptor (GR) under the control of the keratin 5 (K5) promoter (K5-GR mice) exhibited multiple epithelial defects in skin appendages, such as in hair follicle, tooth, and the palate. Importantly, these mice also lacked Meibomian glands [119]. Glucocorticoid receptors can directly bind to NF-κB and, thereby, play a role in regulating NF-κB-driven transcription [120]. These findings, therefore, indicate that NF-κB signaling plays a crucial role in the normal development of ectodermal-derived organs, which could be via the glucocorticoid receptor, thus regulating Meibomian gland development and morphogenesis.

### 7.12. Role of Hyaluronan in Meibomian Gland Development and Homeostasis

Hyaluronan (HA) is a linear polysaccharide formed of a repeating disaccharide unit composed of glucuronic acid and *N*-acetylglucosamine, and is one of the major components in tissues. Our recently published work identified an HA-rich extracellular matrix (ECM) surrounding the basal layer of Meibomian glands [121]. HA is synthesized by HA synthases (HASs), of which mammals have three isoforms: HAS1, HAS2 and HAS3 [121,122]. In our laboratory, *Has* knockout mice were used to unveil the role of HA in Meibomian glands [121]. Adult mice (8 weeks old) lacking *Has1* and *Has3*, hereafter referred to as *Has1^−/−^*; *Has3^−/−^* mice, and conditional inducible mice carrying the tet-O-cre, *K14rtta* and *Has 2^flox/flox^* genes, hereafter referred to as *Has2**^Δ/^**^ΔMG^*, were used to investigate the role of HA in the Meibomian gland. This study demonstrated that *Has1^−/−^;Has3^−/−^* and *Has2**^Δ/^**^ΔMG^* mice present significantly enlarged Meibomian glands when compared to control mice [121]. Moreover, the Meibomian glands of *Has1^−/−^;Has3^−/−^* and *Has2**^Δ/^**^ΔMG^* mice presented stronger Oil-red-O staining, indicating that these glands have increased meibum production when compared to age-matched control mice. Importantly, although *Has1^−/−^, Has3^−/−^* and *Has2**^Δ/^**^ΔMG^* mice lack specific isoforms of the *Has* genes, they up-regulate the remaining isoform(s) in Meibomian glands and tarsal plates, thus the observed phenotype is a result of an increase in HA. This study also found that Meibomian gland progenitor cells and proliferating basal cells reside in an HA-rich niche, and, thus, this niche could provide a supportive role enabling the development of enlarged glands. 

## 8. Discussion

This review outlined the different clinical features of DED and MGD that have been identified in different animal models, including those that have exhibited developmental abnormalities in Meibomian glands (Table 1). There will always be advantages and disadvantages associated with each animal model of choice. Common lab animals, such as rabbits, rats and mice, have all been used to study aspects of DED and MGD, but no single model can completely recapitulate all aspects of clinical MGD. A rabbit model has a longer lifespan with a large exposed ocular surface compared to rodent models, thereby making MGD studies more relevant and also facilitating standard DED clinical tests, such as tear breakup time and fluorescein of the ocular surface. Both dogs and mice have been shown to naturally develop age-related Meibomian gland anatomical changes, similar to those observed in ARMGD. In particular, in collaboration with veterinary clinics, dogs suffering from dry eye are a valuable resource for testing/developing treatments. Due to the vast number of genetic mouse models that currently exist, the mouse model is invaluable for studying the underlying pathologic mechanism of MGD.

## Figures and Tables

**Figure 1 ijms-21-08822-f001:**
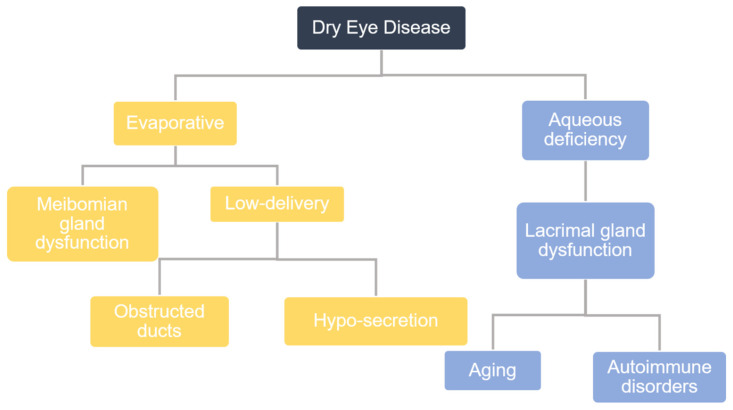
A flowchart representation of the categorization of dry eye disease (DED) according to etiology.

**Table 1 ijms-21-08822-t001:** Transgenic mouse models with Meibomian gland alterations and/or Meibomian gland dysfunction (MGD)-like features *.

Model	Description	Animal	MGD Features	Other Defects
Yagyu, H. et al. 2000.	Elimination of Acyl-CoA:cholesterol acyltransferase gene 1 (ACAT-1).	ACAT-1 *null* (*ACAT-1^−/−^*) mice	Narrow eye fissures and Meibomian gland atrophy.	Lipid-depleted adrenal glands and corneal erosion.
Pikus et al. 2004.	Modulation of bone morphogenic protein (BMP).	K14-Noggin transgenic mouse	Formation of pilosebaceous units at the expense of Meibomian glands/suppression of the induction of Meibomian glands.	Abnormal sweat glands, ectopic cilia, distal limb agenesis, hyperpigmentation of claws, interdigital webbing, and reduced footpads.
Cascallana et al. 2005.	Overexpression of glucocorticoid receptor (GR).	keratin 5 (K5)-GR mice	Lack of Meibomian glands.	Underdeveloped sweat glands and preputial glands, and abnormal hair follicles, teeth, and palate.
Cui, C.Y. et al. 2005; Cui, C.Y. et al. 2010; Wang, Y.C. et al. 2016; Li, S. et al. 2018.	Mutation of X-linked anhidrotic ectodermal dysplasia (EDA) gene.	*Eda* mutant (*Eda^−/−^*) *Tabby* mice	Lack of Meibomian glands, reduced tear break up times, and blepharitis.	Corneal neovascularization, ulceration, keratinization, reduced corneal epithelial microvilli, and conjunctivitis.
Vauclair, S. et al. 2007.	Skin-specific inactivation of the Notch1 gene.	Notch1K14Cre mice; Notch1 K5Cre^ERT^ mice	Meibomian gland dysfunction, abnormal morphology of Meibomian glands, lack of lipids in meibocytes.	Eye irritations, corneal opacity and keratinization.
Chang et al. 2009.	Elevation of EDAR signaling.	*Edar^Tg951^* heterozygous transgenic and *Edar^Tg951/Tg951^* homozygous transgenic mice	Enlarged Meibomian glands.	Excessively branched mammary and salivary glands.
Tukel, T. et al. 2010.	Homozygous *TWIST2* (MIM 607556) nonsense mutations.	*Twist2* knockout (KO) mice	Absence of or hypoplastic Meibomian glands and decreased eyelash follicles.	Bitemporal lesions, narrow snout, pointed chin, and sparse or absent eyelashes.
Kenchegowda, D. et al. 2011.	Conditional deletion of Krüppel-like factor (KLF) 5.	*Klf5*-conditional null (*Klf5*CN) mice	Malformed Meibomian glands with disorganized acini, lipid accumulation in the Meibomian ducts.	Smaller eyeballs; translucent and thicker corneas with a defective epithelial basement membrane and hypercellular stroma; conjunctivas lacking goblet cells.
Tsau, C. et al. 2011.	Deletion of the sequences encoding the homeodomain and C-terminal region of *Barx2*.	*Barx2*-null mice	~50% of *Barx2*^–/–^ mice have eyelid fusion problems, smaller eyes, and defects in Meibomian gland development and structure including Meibomian gland drop out or smaller glands.	Defective lacrimal gland morphogenesis, and absence of the harderian gland.
Lin et al. 2013.	Mice lacking fatty acid transport protein (FATP) 4.	Tg (IVL-Fatp4) transgenic mice	Underdeveloped Meibomian glands with thickened ducts.	Abnormal sebaceous glands, and thick skin with defective barrier.
McMahon et al. 2014.	Mutation in ELOVL4 resulted in abnormal synthesis of extremely long-chain fatty acids.	Heterozygous Stgd3 mice, on a mixed 129SvEv and C57BL6 background	Protruding Meibomian gland orifice, intragland anatomical changes, toothpaste-like meibum, and intense staining for ELOVL4 in glands.	Inability to open eyes fully, and increased blink rates.
Chen, Z.Y. et al. 2014.	Conditional deletion of *Sox9*.	*Sox9^CKO^* mice	Reduced number of Meibomian glands, 40% fewer glands in the upper and lower eyelids, and most Meibomian glands have fewer acini.	Lacrimal gland (LG) deficiency and abnormal LG morphogenesis, absence of harderian glands, and hair loss in mouse eyelids and facial skin.
Meng, Q. et al. 2014.	Targeted gene ablation that inactivated distinct signaling pathways.	*Map3k1*-null and Dkk2-null mice, *K14rtTA/tet-O-cre/Shp2^F/F^* mice, *c-Jun^Δ/ΔOSE^* and *Egfr^Δ/ΔOSE^* mice, *Map3k1/Jnk1* and *Map3k1/Rhoa* compound mutants	Eyelid fusion problems and severe hypoplastic Meibomian glands.	Corneal erosion/ulceration, harderian gland hypoplasia, misplacement of extraocular muscles and eyes open at birth.
Ibrahim, O.M. et al. 2014; Ikeda et al. 2018.	Deficient in Cu/Zn superoxide dismutase (SOD1) leading to accumulation of reactive oxygen species (ROS).	*Sod1*^−/−^ mice	Age-related Meibomian gland abnormalities including an increase in periglandular inflammatory infiltrates, decrease in Meibomian gland glandular acinar density, and increase in periglandular fibrosis.	Corneal fluorescein and lissamine staining evidencing corneal erosions and reduced tear secretion.
Dong, F. et al. 2015.	Conditional inducible ablation of TGFα in the eyelid.	Bi-transgenic *Kera-rtTA/tetO-TGFα (KR/TG)* mice	Abnormal Meibomian glandmorphogenesis, Meibomian gland atrophy, and eyelid tendon and tarsal plate malformation.	Precocious eye opening, swollen eyelids, and conjunctival eyelid epithelial hyperplasia.
Sima, J. et al. 2016; Cui, C.Y. et al. 2010.	Evaluation of Dickkopf 4 (DKK4) regulated signaling pathway.	Skin-specific Dkk4 transgenic mice (WTDk4TG, wildtype background) and TaDk4TG (*Tabby* background)	Meibomian gland formation defects in Dkk4Tg mice, similar with *Tabby* mice.	Cataracts, corneal blindness and rough hair coat.
Miyake et al. 2016.	HR-AD diet (a special diet with limited lipid content).	HR-1 hairless mice (fed with HR-AD diet)	Plugged Meibomian gland orifice, Meibomian gland ductal epithelial hyperkeratinization and acinar atrophy.	No other phenotypes recorded.
Reneker, L.W. et al. 2017; Chen, Z.Y. et al. 2014.	Conditional inducible ablation of FGFR2.	Inducible *Fgfr2^CKO^* mice	Macerated periorbital hairs and eyelids, Meibomian gland orifice obstruction, reduced volume of meibum and Meibomian gland acinar atrophy.	Ocular irritation and rubbing.
Yu, D. et al. 2018.	Deletion of epithelial sodium channel (ENaC) functional subunits in the Meibomian gland.	Conditional *βENaC* Meibomian gland knockout (KO) mouse (*βENaC*^fl/fl^;Shh-Cre^+/−^)	Age-dependent, female-predominant Meibomian gland dysfunction (obstruction of Meibomian gland orifices and Meibomian gland acinar atrophy).	Increased tear secretion and severe ocular surface damage (corneal opacity, ulceration, neovascularization and ectasia).
Swirski, S. et al. 2018.	Mutations in the gene encoding Gasdermin A3 (*Gsdma3*).	C+/H− Mice	Degeneration of Meibomian glands, increased eyelid tissue formation and eyelid closure.	Progressive hair loss, hyperkeratinization of the skin, degeneration of sebaceous glands, corneal opacity, and corenal vascularization.
Sassa, T. et al. 2018.	Gene disruption of Elovl1 resulted in insufficient elongation of FAs.	*Elovl1^−/−^ Tg(IVL-Elovl1)* mice	Evaporative dry eye phenotype with increased eye-blink frequency, together with partially closed eyes and excessive tear production.	Corneal opacity, vascularization and epidermalization in aged mice.
Sun, M. et al. 2018; Gesteira, T.F. et al. 2017.	Targeted knockout of Hyaluronan synthase (Has) genes 1, 2 and 3.	Combined *Has1^−/−^*;*Has3^−/−^* mice; conditional *Has2^Δ/ΔMG^* mice	Meibomian gland hyperplasia with more branched, longer and wider acini.	Dysmorphic eyelids with conjunctival epithelial hyperplasia and thinner corneal thickness.
Swirski, S. et al. 2018; Miyazaki, M. et al. 2001.	Targeted knockout of Stearoyl-CoA desaturase (SCD) 1.	SCD-1 knockout mice (*SCD-1**^−/−^*)	Narrow eye fissures, atrophy or loss of Meibomian glands and depletion of meibum lipids.	Cutaneous abnormalities including thinner hair coat and hair loss, with atrophic sebaceous gland abnormalities and compensatory increases in tear volume and mucin levels.
Sundberg, J.P. et al. 2018.	*Null* mice for fatty acyl CoA reductase 2 (*Far2*) gene altering saturated 16:0 and 18:0 carbon fatty acids.	Homozygous mutant *Far2^tm2b(KOMP)Wtsi^*/2J (*Far2**^−/−^*; *null*)	Markedly dilated Meibomian gland ducts, and abnormal sebocytes with clustered eosinophilic remnants.	Patchy alopecia on the dorsal trunk and diffuse hair thinning on the ventral body surface.
Butovich, I.A. et al. 2019.	Loss of ELOVL3 altering the synthesis of C21:0-C29:0 fatty acids.	*Elovl3* knockout (*E3hom*) mice	Delayed eye opening, weeping eyes, crusty eyelids, eyelid edema, highly vascularized cornea and tarsal plates (TPs), slit eye, and excessive tearing.	Hairless pups and greasy fur in adults.
Wu, L. et al. 2019.	Null SDR16C5 and SDR16C6 mice with altered retinoic acid (RA) biosynthesis.	*Sdr16c5/Sdr16c6* double-knockout mice (DKO)	Both the upper and lower eyelids are thicker and longer (eyelid expansion) with larger Meibomian glands.	Accelerated hair regrowth.
Widjaja-Adhi, M.A.K. et al. 2020.	Ablation of Acyl-CoA wax alcohol acyltransferase 2 (AWAT2) enzymatic activity.	Awat2-knockout mice (*Awat2^−/−^)*	Obstruction of the orifice and excretory duct of Meibomian glands, hyperkeratinization of the epithelium, absence of wax esters and overproduction of cholesteryl esters.	Shorter tear film break-up time, deterioration of the corneal surface, granular deposits in the corneal stromal layer, corneal neovascularization, blepharitis, Iritis, scaly and dry skin, and smaller sebaceous glands.

* This table summarizes findings from studies published between 2000 and 2020, which have been ordered chronologically.

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
