# Peer review of "Meibomian Gland Dysfunction: What Have Animal Models Taught Us?"

_ijms, 2020, doi:10.3390/ijms21228822_

Round 1

Reviewer 1 Report

the manuscript is well structured however it needs some modifications to be more publishable.
the chapter of the introduction must be extended with specifics to the animal models studied for the different pathologies. the chapter on materials and methods needs to be simplified slightly.
in the paragraph (7.9. FGFR2 signaling in Meibomian gland homeostasis and potential role in MGD) I suggest to add some bibliographic notes related to the fgf.(Scalinci SZ, Scorolli L, Meduri A, Grenga PL, Corradetti G, Metrangolo C. Effect of basic fibroblast growth factor and cytochrome c peroxidase combination in transgenic mice corneal epithelial healing process after excimer laser photoablation. Clin Ophthalmol. 2011;5:215-21. doi: 10.2147/OPTH.S16866. Epub 2011 Feb 16. PMID: 21386914; PMCID: PMC3046991.

Meduri A, Scalinci SZ, Morara M, Ceruti P, Grenga PL, Zigiotti GL, Scorolli L. Effect of basic fibroblast growth factor in transgenic mice: corneal epithelial healing process after excimer laser photoablation. Ophthalmologica. 2009;223(2):139-44. doi: 10.1159/000187686. Epub 2008 Dec 18. PMID: 19092284.

Author Response

First of all, we would like to thank the editor and the reviewers for their time and feedback on our manuscript. We hereby provide a point-by-point response to the issues raised by the reviewers.

Reviewer 1

the manuscript is well structured however it needs some modifications to be more publishable.

the chapter of the introduction must be extended with specifics to the animal models studied for the different pathologies. the chapter on materials and methods needs to be simplified slightly.

  1. We have provided some specifics to the animal models for the different pathologies in the introduction, as requested. Unfortunately, since we do not have a “materials and methods” section we do not really understand what has been requested and we have therefore been unable to make the requested change.

in the paragraph (7.9. FGFR2 signaling in Meibomian gland homeostasis and potential role in MGD) I suggest to add some bibliographic notes related to the fgf.(Scalinci SZ, Scorolli L, Meduri A, Grenga PL, Corradetti G, Metrangolo C. Effect of basic fibroblast growth factor and cytochrome c peroxidase combination in transgenic mice corneal epithelial healing process after excimer laser photoablation. Clin Ophthalmol. 2011;5:215-21. doi: 10.2147/OPTH.S16866. Epub 2011 Feb 16. PMID: 21386914; PMCID: PMC3046991.

Meduri A, Scalinci SZ, Morara M, Ceruti P, Grenga PL, Zigiotti GL, Scorolli L. Effect of basic fibroblast growth factor in transgenic mice: corneal epithelial healing process after excimer laser photoablation. Ophthalmologica. 2009;223(2):139-44. doi: 10.1159/000187686. Epub 2008 Dec 18. PMID: 19092284.

  1. These references have been added.

  1. Please note that since the reviewer selected “Moderate English changes required” we have also carefully proofread the manuscript.

Reviewer 2 Report

This paper provides an extensive review of the current knowledge of MGD and how animal models inform research on humans and their importance in identifying pathomechanisms leading to MGD.

This is the first review summarizing findings based on animal model MGD and their  application and usefulness for human MGD understanding and research.
The review is well structured and written. MGD is affecting a large proportion of the population and with increasing life expectancy combined with more people wearing contact lenses there will be more  patients in the future requiring clinical assessment and possible treatment,  hence the topic is of importance to both researchers and clinicians.

Author Response

First of all, we would like to thank the editor and the reviewers for their time and feedback on our manuscript. We hereby provide a point-by-point response to the issues raised by the reviewers.

Reviewer 2

This paper provides an extensive review of the current knowledge of MGD and how animal models inform research on humans and their importance in identifying pathomechanisms leading to MGD.

This is the first review summarizing findings based on animal model MGD and their  application and usefulness for human MGD understanding and research.

The review is well structured and written. MGD is affecting a large proportion of the population and with increasing life expectancy combined with more people wearing contact lenses there will be more  patients in the future requiring clinical assessment and possible treatment,  hence the topic is of importance to both researchers and clinicians.

  1. We thank the reviewer for his comments. We have carefully proofread the manuscript since the reviewer selected the option

“English language and style are fine/minor spell check required”.